# PHASED DMD: FEW-STEP DISTRIBUTION MATCHING DISTILLATION VIA SCORE MATCHING WITHIN SUBINTERVALS

## ABSTRACT

Distribution Matching Distillation (DMD) distills score-based generative models into efficient one-step generators, without requiring a one-to-one correspondence with the sampling trajectories of their teachers. However, limited model capacity causes one-step distilled models underperform on complex generative tasks, *e.g.*, synthesizing intricate object motions in text-to-video generation. Directly extending DMD to multi-step distillation increases memory usage and computational depth, leading to instability and reduced efficiency. While prior works propose stochastic gradient truncation as a potential solution, we observe that it substantially reduces the generation diversity of multi-step distilled models, bringing it down to the level of their one-step counterparts. To address these limitations, we propose **Phased DMD**, a multi-step distillation framework that bridges the idea of phase-wise distillation with Mixture-of-Experts (MoE), reducing learning difficulty while enhancing model capacity. Phased DMD is built upon two key ideas: *progressive distribution matching* and *score matching within subintervals*. First, our model divides the SNR range into subintervals, progressively refining the model to higher SNR levels, to better capture complex distributions. Next, to ensure the training objective within each subinterval is accurate, we have conducted rigorous mathematical derivations. We validate Phased DMD by distilling state-of-the-art (SOTA) image and video generation models, including Qwen-Image (20B parameters) and Wan2.2 (28B parameters). Experimental results demonstrate that Phased DMD preserves output diversity better than DMD while retaining key generative capabilities. We will release our code and models.

## 1 INTRODUCTION

Recently, state-of-the-art (SOTA) diffusion models have made significant progress in image and video generation. In image generation, SOTA models (Wu et al., 2025; OpenAI, 2025; Team, 2025b; GoogleAI, 2025a) demonstrate precise prompt control, enabling complex text-to-image rendering and accurate layout specification. In video generation, these models (Wan et al., 2025; Kong et al., 2024; GoogleAI, 2025b; OpenAI, 2024) exhibit substantial improvements in dynamic scene generation, such as fast-moving objects in sports and complex camera movements like ego-centric videos. Simultaneously, the increasing parameter sizes and computational demands of base models highlight the importance of accelerating diffusion model sampling.

Several techniques have been proposed to accelerate diffusion models, including classifier-free guidance (CFG) distillation (Meng et al., 2023), step distillation (Song et al., 2023; Wang et al., 2024; Salimans & Ho, 2022; Yin et al., 2024a; Luo et al., 2023; Luo, 2024; Zhou et al., 2024; Huang et al., 2024; Lin et al., 2024; 2025; Frans et al., 2024; Geng et al., 2025), SVDQuant (Li* et al., 2025), Mixture-of-Expert (MoE) models (Balaji et al., 2022; Feng et al., 2023; Wan et al., 2025), and parallel computation (Fang et al., 2024). Among these, step distillation methods based on Variational Score Distillation(VSD), including diff-instruct (Luo et al., 2023), DMD (Yin et al., 2024a), SID (Zhou et al., 2024), achieve high-quality generation by distilling models into single-step generators. However, the limited network capacity (Lin et al., 2024) of single-step distilled models hinders their ability to handle complex tasks like intricate text rendering or dynamic scene generation, which are critical for the widespread adoption of these foundational models.

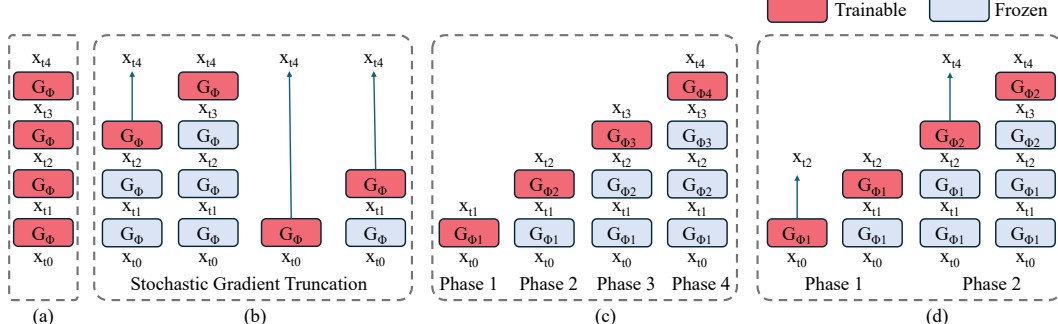

Figure 1: Schematic diagram of (a) Few-step DMD (Yin et al., 2024a), (b) Few-step DMD with stochastic gradient truncation strategy (SGTS) (Huang et al., 2025), (c) Phased DMD and (d) Phased DMD with SGTS .

Few-step distillation balances computational cost and generation quality (Luo et al., 2025). Yet, as shown in Fig. 1a, directly applying VSD to few-step distillation (Yin et al., 2024a) introduces challenges such as increased computational graph depth and higher memory overhead. Furthermore, the lack of explicit constraints on intermediate generator steps reduces training stability and leads to suboptimal performance in few-step models. To address these issues, Huang et al. (2025) proposed a stochastic gradient truncation strategy (SGTS), where multi-step sampling may terminate at a random step and the gradient backpropagation is restricted to the final denoising step (see Fig. 1b). This approach improves training convergence and stability by supervising all intermediate steps while enhancing memory efficiency via gradient detachment for non-final steps. However, SGTS can terminate sampling after just one step during training, distilling a one-step generator for that iteration. Consequently, the generative diversity of few-step generators trained with SGTS is reduced to a level akin to that of one-step generators.

The diffusion theory (Song et al., 2020) suggests the existence of infinitely many neural networks as score estimators across a range of signal-to-noise ratios (SNR), spanning from zero to infinity. During the generation process, diffusion models exhibit distinct temporal dynamics (Balaji et al., 2022). Specifically, the low-SNR stage focuses on modeling image structures and video dynamics, while the high-SNR stage refines visual details. In practice, a single neural network is typically employed throughout the denoising process, requiring the model to simultaneously learn and perform a variety of denoising tasks. Recent studies (Balaji et al., 2022; Feng et al., 2023; Wan et al., 2025) have introduced an MoE architecture into diffusion models. By assigning specialized experts to different SNR levels, MoE enhances model capacity and generative performance without increasing inference cost. The performance improvement is particularly pronounced in video generation (Wan et al., 2025), where the low-SNR expert excels at capturing dynamic content.

In this work, we propose **Phased DMD**, a novel distillation framework for few-step generation. Our approach is inspired by a broader vision: *By decomposing a complex task into learnable phases, each phase naturally forms an expert, collectively enhancing the model's capacity in a MoE manner.* Our method is built upon two key components:

- **Progressive distribution matching:** Conceptually similar to ProGAN (Karras et al., 2017), which progressively trains a generator to handle higher resolutions, Phased DMD divides SNR into subintervals and progressively distills models toward higher SNR levels.

- **Score matching within SNR subintervals:** As each phase is trained within a subinterval, the training objective undergoes a transformation. To ensure theoretical rigor, we derive the training objective for the fake score estimator within each subinterval.

As illustrated in Fig. 1c, Phased DMD offers several advantages: **First**, by partitioning SNR into subintervals, the model learns complex data distributions incrementally, improving training stability and generative performance. **Second**, each phase involves only a single gradient-recorded sampling step, avoiding additional computational and memory overhead. **Third**, notably, Phased DMD naturally produces a few-step MoE generative model, regardless of whether the teacher model adopts an

MoE architecture. **Last**, as shown in Fig. 1d, Phased DMD can be combined with SGTS , enabling 4-step inference across 2 phases while simplifying the complexity of both training and inference.

We validate Phased DMD by distilling SOTA image and video generation models, including Qwen-Image (Wu et al., 2025) with 20B parameters and Wan2.1/Wan2.2 (Wan et al., 2025) with 14/28B parameters. Experimental results demonstrate that Phased DMD better preserves output diversity compared to standard DMD while maintaining the base models' key capabilities, such as faithful text rendering in Qwen-Image and realistic dynamic motion in Wan2.2.

Our contributions are summarized as follows:

- We propose Phased DMD, a data-free distillation framework for few-step diffusion models. This framework combines ideas from DMD and MoE, achieving higher performance ceilings while maintaining memory usage similar to single-step distillation.
- We derive the theoretical training objective for subinterval diffusion models without relying on external information, such as clean samples. We highlight the necessity of this correctness for DMD distillation.
- Without requiring GAN loss or regression loss, Phased DMD achieves SOTA results on text-to-image and text-to-video generation models. To the best of our knowledge, this is the largest reported distillation validation. Experimental results show that our method effectively reduces diversity loss while preserving the base models' key capabilities, including complex text rendering and high-dynamic video generation.

## 2 METHOD

To clarify the principle of phased DMD, we begin by introducing the theoretical background and notations related to diffusion models (Kingma et al., 2023), score matching (Song et al., 2020; Karras et al., 2022), and distribution matching distillation (Yin et al., 2024b;a). We explicitly highlight why the principle of DMD is applicable only to score-based generative models. Building on this foundation, we present the motivation behind Phased DMD and explain how it inherently achieves improved generative diversity. Following this, we detail the two key components of Phased DMD : progressive distribution matching and score matching within subintervals.

### 2.1 PRELIMINARY

#### 2.1.1 DIFFUSION MODELS AND SCORE MATCHING

Consider a continuous-time Gaussian diffusion process defined over the interval $0 \leq t \leq 1$. The ground-truth distribution is denoted $p(\boldsymbol{x}_0)$. For any $0 \leq t \leq 1$, the forward diffusion process is described by the following conditional distribution:

$$p(\boldsymbol{x}_t|\boldsymbol{x}_0) = \mathcal{N}(\boldsymbol{x}_t; \alpha_t \boldsymbol{x}_0, \sigma_t^2 \boldsymbol{I}) \tag{1}$$

where $\alpha_t$ and $\sigma_t^2$ are positive, scalar-valued functions of $t$. The signal-to-noise ratio (SNR) is defined as $\text{SNR}(t) = \alpha_t^2/\sigma_t^2$. It is assumed that $\text{SNR}(t)$ is strictly monotonically decreasing over time. No additional constraints are imposed on the relationship between $\alpha_t$ and $\sigma_t$, ensuring the notations are compatible with different kinds of diffusion models (Ho et al., 2020; Karras et al., 2022; Song et al., 2022; Podell et al., 2023) and flow models (Liu et al., 2022; Esser et al., 2024). The diffusion process is Markovian (Kingma et al., 2023), meaning that $p(\boldsymbol{x}_t|\boldsymbol{x}_s, \boldsymbol{x}_0) = p(\boldsymbol{x}_t|\boldsymbol{x}_s)$. Furthermore, $p(\boldsymbol{x}_t|\boldsymbol{x}_s)$ is also Gaussian, and can be expressed as:

$$p(\boldsymbol{x}_t|\boldsymbol{x}_s) = \mathcal{N}(\boldsymbol{x}_t; \alpha_{t|s} \boldsymbol{x}_s, \sigma_{t|s}^2 \boldsymbol{I}) \tag{2}$$

where $\alpha_{t|s} = \alpha_t/\alpha_s$ and $\sigma_{t|s}^2 = \sigma_t^2 - \alpha_{t|s}^2 \sigma_s^2$. For any $0 \leq s < t \leq 1$, the marginal distribution of $\boldsymbol{x}_s$ and $\boldsymbol{x}_t$ are given by $p(\boldsymbol{x}_s) = \int p(\boldsymbol{x}_s|\boldsymbol{x}_0)p(\boldsymbol{x}_0)d\boldsymbol{x}_0$ and $p(\boldsymbol{x}_t) = \int p(\boldsymbol{x}_t|\boldsymbol{x}_0)p(\boldsymbol{x}_0)d\boldsymbol{x}_0$. If only $p(\boldsymbol{x}_s)$ is observed and not $p(\boldsymbol{x}_0)$, the marginal distribution of $\boldsymbol{x}_t$ can alternatively be expressed as: $p(\boldsymbol{x}_t) = \int p(\boldsymbol{x}_t|\boldsymbol{x}_s)p(\boldsymbol{x}_s)d\boldsymbol{x}_s$. Thus, we have the following equivalence:

$$p(\boldsymbol{x}_t) = \int p(\boldsymbol{x}_t|\boldsymbol{x}_0)p(\boldsymbol{x}_0)d\boldsymbol{x}_0 = \int p(\boldsymbol{x}_t|\boldsymbol{x}_s)p(\boldsymbol{x}_s)d\boldsymbol{x}_s \tag{3}$$

In the training process, $\alpha_t$ and $\sigma_t$ are predefined functions of $t$, while $\boldsymbol{x}_0$ is sampled from the dataset distribution $\boldsymbol{x}_0 \sim p(\boldsymbol{x}_0)$. Timestep $t$ is sampled from a predefined distribution over the interval $[0, 1]$, such as a uniform or logit-normal distribution (Esser et al., 2024), i.e., $t \sim \mathcal{T}(t; 0, 1)$. The sample $\boldsymbol{x}_t$ is then given by $\boldsymbol{x}_t = \alpha_t \boldsymbol{x}_0 + \sigma_t \boldsymbol{\epsilon}$, where $\boldsymbol{\epsilon} \sim \mathcal{N}(\boldsymbol{\epsilon}; \mathbf{0}, \boldsymbol{I})$. We use $t \sim \mathcal{T}$ and $\boldsymbol{\epsilon} \sim \mathcal{N}$ for brevity in later paragraphs unless otherwise specified. Song et al. (2020) unified diffusion models under the theoretical framework of score-based generative models and demonstrated that the continuous diffusion process is fundamentally governed by a Stochastic Differential Equation (SDE). Here, we adopt flow velocity prediction as an example and demonstrate its connection to score matching. Let $\psi_{\boldsymbol{\theta}}$ denote a diffusion model parameterized by $\boldsymbol{\theta}$. The relationship between flow matching and score matching is expressed below.

$$J_{flow}(\boldsymbol{\theta}) = \mathbb{E}_{\boldsymbol{x}_0 \sim p(\boldsymbol{x}_0), \boldsymbol{\epsilon} \sim \mathcal{N}, t \sim \mathcal{T}, \boldsymbol{x}_t = \alpha_t \boldsymbol{x}_0 + \sigma_t \boldsymbol{\epsilon}}[\|\psi_{\boldsymbol{\theta}}(\boldsymbol{x}_t) - (\boldsymbol{\epsilon} - \boldsymbol{x}_0)\|^2] \tag{4}$$

$$= \mathbb{E}_{\boldsymbol{x}_0 \sim p(\boldsymbol{x}_0), t \sim \mathcal{T}, \boldsymbol{x}_t \sim p(\boldsymbol{x}_t|\boldsymbol{x}_0)}[\|\psi_{\boldsymbol{\theta}}(\boldsymbol{x}_t) + \boldsymbol{x}_t/\alpha_t + (\sigma_t + \sigma_t^2/\alpha_t)\nabla \boldsymbol{x}_t \log(p(\boldsymbol{x}_t|\boldsymbol{x}_0))\|^2]$$

$$= \mathbb{E}_{t \sim \mathcal{T}, \boldsymbol{x}_t \sim p(\boldsymbol{x}_t)}[\|\psi_{\boldsymbol{\theta}}(\boldsymbol{x}_t) + \boldsymbol{x}_t/\alpha_t + (\sigma_t + \sigma_t^2/\alpha_t)\nabla \boldsymbol{x}_t \log(p(\boldsymbol{x}_t))\|^2] \tag{5}$$

Eq. 5 is derived based on the equivalence between denoising score matching (DSM) and explicit score matching (ESM), as originally proven in Vincent (2011). In Supp. A, we provide the detailed derivation of Eq. 5. Additionally, we demonstrate the connection between sample prediction (a.k.a. x-prediction) and score matching in Appendix A.

### 2.1.2 DISTRIBUTION MATCHING DISTILLATION

Let $\boldsymbol{G}_{\boldsymbol{\phi}}$ denote the generator parameterized by $\boldsymbol{\phi}$. The objective of DMD is to minimize the reverse Kullback-Leibler (KL) divergence between the real data distribution $p_{real}(\boldsymbol{x}_0)$ and the generated data distribution $p_{fake}(\boldsymbol{x}_0)$, produced by $\boldsymbol{G}_{\boldsymbol{\phi}}$.

$$D_{KL}(p_{fake}\|p_{real}) = \mathbb{E}_{\boldsymbol{\epsilon} \sim \mathcal{N}, \boldsymbol{x}_0 = \boldsymbol{G}_{\boldsymbol{\phi}}(\boldsymbol{\epsilon})}[\log p_{fake}(\boldsymbol{x}_0) - \log p_{real}(\boldsymbol{x}_0)] \tag{6}$$

We use $D_{KL}$ to abbreviate $D_{KL}(p_{fake}\|p_{real})$ in later paragraphs. To leverage the pretrained diffusion models as score estimators, the generated samples are diffused and the objective becomes:

$$D_{KL} = \mathbb{E}_{\boldsymbol{\epsilon} \sim \mathcal{N}, \boldsymbol{x}_0 = \boldsymbol{G}_{\boldsymbol{\phi}}(\boldsymbol{\epsilon}), t \sim \mathcal{T}, \boldsymbol{x}_t \sim p(\boldsymbol{x}_t|\boldsymbol{x}_0)}[\log p_{fake}(\boldsymbol{x}_t) - \log p_{real}(\boldsymbol{x}_t)] \tag{7}$$

By combining Eq. 5 and Eq. 7, we can approximate the objective as:

$$D_{KL} \approx \mathbb{E}_{\boldsymbol{\epsilon} \sim \mathcal{N}, \boldsymbol{x}_0 = \boldsymbol{G}_{\boldsymbol{\phi}}(\boldsymbol{\epsilon}), t \sim \mathcal{T}, \boldsymbol{x}_t \sim p(\boldsymbol{x}_t|\boldsymbol{x}_0)}[\lambda_t(\boldsymbol{T}_{\hat{\boldsymbol{\theta}}}(\boldsymbol{x}_t) - \boldsymbol{F}_{\boldsymbol{\theta}}(\boldsymbol{x}_t))] \tag{8}$$

where $\lambda_t = 1/(\sigma_t + \sigma_t^2/\alpha_t)$, $\boldsymbol{F}_{\boldsymbol{\theta}}$ denotes the fake diffusion model and $\boldsymbol{T}_{\hat{\boldsymbol{\theta}}}$ denotes the teacher diffusion model. $\boldsymbol{\theta}$ is initialized from $\hat{\boldsymbol{\theta}}$ and $\boldsymbol{F}_{\boldsymbol{\theta}}$ is updated on $p_{fake}(\boldsymbol{x}_0)$ according to Eq. 4. The derivation from Eq. 7 to Eq. 8 is valid under the condition that the models are score-based generative models. Formally, this approximation holds if $\boldsymbol{F}_{\boldsymbol{\theta}}(\boldsymbol{x}_t) \approx a_t \nabla \boldsymbol{x}_t \log(p_{fake}(\boldsymbol{x}_t)) + b_t \boldsymbol{x}_t$ and $\boldsymbol{T}_{\hat{\boldsymbol{\theta}}}(\boldsymbol{x}_t) \approx a_t \nabla \boldsymbol{x}_t \log(p_{real}(\boldsymbol{x}_t)) + b_t \boldsymbol{x}_t$, where $a_t$ is any non-zero function of $t$ and $b_t$ is any function of $t$. Taking the gradient of Eq. 8 with respect to the generator parameters, we have:

$$\nabla \boldsymbol{\phi} D_{KL} \approx \mathbb{E}_{\boldsymbol{\epsilon} \sim \mathcal{N}, \boldsymbol{x}_0 = \boldsymbol{G}_{\boldsymbol{\phi}}(\boldsymbol{\epsilon}), t \sim \mathcal{T}, \boldsymbol{x}_t \sim p(\boldsymbol{x}_t|\boldsymbol{x}_0)}[w_t(\boldsymbol{T}_{\hat{\boldsymbol{\theta}}}(\boldsymbol{x}_t) - \boldsymbol{F}_{\boldsymbol{\theta}}(\boldsymbol{x}_t))]d\boldsymbol{G}/d\boldsymbol{\phi} \tag{9}$$

where $w_t = \lambda_t \alpha_t$. Similar to GANs (Goodfellow et al., 2014), DMD employs an adversarial training process consisting of two stages in each iteration. In the fake diffusion optimization stage, $\boldsymbol{F}_{\boldsymbol{\theta}}$ is optimized on the generated distribution using Eq. 4, allowing it to serve as a score estimator for $p_{fake}(\boldsymbol{x}_t)$. In the generator optimization stage, $\boldsymbol{G}_{\boldsymbol{\phi}}$ is updated according to Eq. 9, encouraging the generated distribution to more closely approximate the real distribution. For training stability, $\boldsymbol{F}_{\boldsymbol{\theta}}$ receives more frequent updates, enabling it to accurately estimate the score of the evolving generated distribution (Yin et al., 2024a).

### 2.2 FROM ONE-STEP DISTILLATION TO FEW-STEP DISTILLATION

In $N$-step distillation, we have a scheduler $\mathcal{S}$ with $N + 1$ timesteps, $\boldsymbol{t} = \{t_0, t_1, t_2, ..., t_N\}$, where $0 = t_N < t_i < t_{i-1} < t_0 = 1$ for any $i \in \{2, ..., N - 1\}$. The sampling process begins with $\boldsymbol{x}_{t_0} = \boldsymbol{\epsilon} \sim \mathcal{N}(\boldsymbol{\epsilon}; \mathbf{0}, \boldsymbol{I})$. The sample $\boldsymbol{x}_0$ is then generated iteratively: for $i = 0, 1, ..., N - 1$, we compute $\boldsymbol{x}_{t_{i+1}} = \mathcal{S}(\boldsymbol{G}_{\boldsymbol{\phi}}(\boldsymbol{x}_{t_i}), \boldsymbol{x}_{t_i}, t_i, t_{i+1})$. Let $\text{pipeline}(\boldsymbol{G}_{\boldsymbol{\phi}}, \boldsymbol{t}, \boldsymbol{\epsilon}, \mathcal{S})$ denote this iterative sampling procedure. Eq. 9 is thus adapted as follows:

$$\nabla \boldsymbol{\phi} D_{KL} \approx \mathbb{E}_{\boldsymbol{\epsilon} \sim \mathcal{N}, \boldsymbol{x}_0 = \text{pipeline}(\boldsymbol{G}_{\boldsymbol{\phi}}, \boldsymbol{t}, \boldsymbol{\epsilon}, \mathcal{S}), t \sim \mathcal{T}, \boldsymbol{x}_t \sim p(\boldsymbol{x}_t|\boldsymbol{x}_0)}[w_t(\boldsymbol{T}_{\hat{\boldsymbol{\theta}}}(\boldsymbol{x}_t) - \boldsymbol{F}_{\boldsymbol{\theta}}(\boldsymbol{x}_t))]d\boldsymbol{G}/d\boldsymbol{\phi} \tag{10}$$

As shown in Fig. 1a, the depth of the computational graph during generator optimization increases linearly with $N$, which reduces training stability and increases memory overhead. To address this issue, Huang et al. (2025) introduced a stochastic gradient truncation strategy (SGTS), depicted in Fig. 1b. In this strategy, an index $j$ is randomly selected from $\{1, 2, ..., N\}$, the corresponding timestep $t_j$ is set to 0. The sampling pipeline is then executed only for steps $i = 0, 1, ..., j - 1$. Crucially, when $j = 1$, the training iteration reduces to a one-step distillation. Consequently, while SGTS improves memory efficiency and training stability, it reduces the generative diversity of the few-step models, as the generated distribution is biased toward that of a one-step generator.

## 2.3 PHASED DMD

In contrast to DMD with SGTS, which can degenerate into one-step distillation in certain iterations, Phased DMD avoids this issue by partitioning the distillation process into distinct phases and applying supervision at intermediate timesteps. In each phase except the last, the generator is optimized to minimize the reverse KL divergence at an intermediate timestep, while the fake diffusion model is updated via score matching within a subinterval of the diffusion process.

### 2.3.1 DISTRIBUTION MATCHING AT INTERMEDIATE TIMESTEPS

The motivation for Phased DMD can be understood by revisiting Eq. 10. To sample $\boldsymbol{x}_t$, prior methods (Yin et al., 2024a; Huang et al., 2025) first generate $\boldsymbol{x}_0$ and then diffuse it to $\boldsymbol{x}_t$ according to Eq. 1. In phased DMD, the pipeline is modified to generate intermediate samples $\boldsymbol{x}_{t_k}$, where $0 < k \le N$, instead of $\boldsymbol{x}_0$. The sample $\boldsymbol{x}_{t_k}$ is then diffused according to Eq. 2, with $s = t_k$ and $t$ is sampled from the subinterval $(t_k, 1)$, i.e., $t \sim \mathcal{T}(t; t_k, 1)$. As illustrated in Fig. 1c, Phased DMD progressively distills the generator toward higher SNR levels. In each phase $k$, only a single expert $\boldsymbol{G}_{\boldsymbol{\phi}_k}$ is trained. This expert maps the distribution $p(\boldsymbol{x}_{t_{k-1}})$ to $p(\boldsymbol{x}_{t_k})$. The generator optimization objective for the $k$-th phase is given by:

$$\nabla \boldsymbol{\phi_k} D_{KL} \approx \mathbb{E}_{\boldsymbol{\epsilon} \sim \mathcal{N}, \boldsymbol{x}_{t_k} = \text{pipeline}(\boldsymbol{G}_{\boldsymbol{\phi_1}}, \boldsymbol{G}_{\boldsymbol{\phi_2}}, ..., \boldsymbol{G}_{\boldsymbol{\phi_k}}, \{t_1, t_2, ..., t_k\}, \boldsymbol{\epsilon}, \mathcal{S}), t \sim \mathcal{T}(t; t_k, 1), \boldsymbol{x}_t \sim p(\boldsymbol{x}_t | \boldsymbol{x}_{t_k})}$$
$$[w_t(\boldsymbol{T}_{\hat{\boldsymbol{\theta}}}(\boldsymbol{x}_t) - \boldsymbol{F}_{\boldsymbol{\theta}_i}(\boldsymbol{x}_t))] d\boldsymbol{G}/d\boldsymbol{\phi_k} \tag{11}$$

Empirically, we find that sampling $t \sim \mathcal{T}(t; t_k, 1)$ instead of $t \sim \mathcal{T}(t; t_k, t_{k-1})$, aligns better with the progressive design of Phased DMD and yields superior performance. At the onset of each phase, the fake diffusion model $\boldsymbol{F}_{\boldsymbol{\theta}_k}$ is re-initialized from the pretrained teacher model $\boldsymbol{T}_{\hat{\boldsymbol{\theta}}}$ and is trained independently of the models from previous phases.

Although the resulting MoE generator requires more GPU memory than a single-network generator, the overhead is manageable for three reasons. First, an optimizer is required only for the $k$-th trainable expert. Second, this overhead can be substantially reduced using Low-Rank Adaptation (LoRA) (Hu et al., 2021). Specifically, all experts can share a common backbone network, with individual experts activated by switching their respective LoRA weights. Finally, Phased DMD can be combined with SGTS (as shown in Fig. 1d), and the number of distillation phases can be less than the number of sampling steps.

### 2.3.2 SCORE MATCHING WITHIN SUBINTERVALS

A key challenge in Phased DMD is that clean data samples $\boldsymbol{x}_0$ are inaccessible in all but the final phase. Consequently, the training objective for the fake diffusion model $\boldsymbol{F}_{\boldsymbol{\theta}_k}$ in Eq. 4 is no longer applicable. To address this, we derive a training objective based on score matching within subintervals. Assume we have observations $\boldsymbol{x}_s \sim p(\boldsymbol{x}_s)$ at an intermediate timestep $s$ where $0 < s < 1$. The diffusion model $\boldsymbol{\psi}_{\boldsymbol{\theta}}$ can be optimized within the subinterval $(s, 1)$ using the following objective, derived from Eq. 5:

$$J_{flow}(\boldsymbol{\theta}) = \mathbb{E}_{t \sim \mathcal{T}(t;s,1), \boldsymbol{x}_t \sim p(\boldsymbol{x}_t)}[\|\boldsymbol{\psi}_{\boldsymbol{\theta}}(\boldsymbol{x}_t) + \boldsymbol{x}_t/\alpha_t + (\sigma_t + \sigma_t^2/\alpha_t)\nabla \boldsymbol{x}_t \log(p(\boldsymbol{x}_t))\|^2]$$

$$= \mathbb{E}_{\boldsymbol{x}_s \sim p(\boldsymbol{x}_s), t \sim \mathcal{T}(t;s,1), \boldsymbol{x}_t \sim p(\boldsymbol{x}_t | \boldsymbol{x}_s)}[\|\boldsymbol{\psi}_{\boldsymbol{\theta}}(\boldsymbol{x}_t) + \boldsymbol{x}_t/\alpha_t + (\sigma_t + \sigma_t^2/\alpha_t)\nabla \boldsymbol{x}_t \log(p(\boldsymbol{x}_t | \boldsymbol{x}_s))\|^2]$$

$$= \mathbb{E}_{\boldsymbol{x}_s \sim p(\boldsymbol{x}_s), \boldsymbol{\epsilon} \sim \mathcal{N}, t \sim \mathcal{T}(t;s,1), \boldsymbol{x}_t = \alpha_{t|s}\boldsymbol{x}_s + \sigma_{t|s}\boldsymbol{\epsilon}}[\|\boldsymbol{\psi}_{\boldsymbol{\theta}}(\boldsymbol{x}_t) - ((\alpha_s^2\sigma_t + \alpha_t\sigma_s^2)/(\alpha_s^2\sigma_{t|s})\boldsymbol{\epsilon} - (1/\alpha_s)\boldsymbol{x}_s)\|^2] \tag{12}$$

In the $k$-th phase of Phased DMD, the distribution $p(\boldsymbol{x}_s)$ is approximated using the output of the MoE generator pipeline $\boldsymbol{G}_{\boldsymbol{\phi_1}}, \boldsymbol{G}_{\boldsymbol{\phi_2}}, ..., \boldsymbol{G}_{\boldsymbol{\phi_k}}$. As $\sigma_{t|s} \to 0$ when $t \to s$, the formulation in Eq. 12

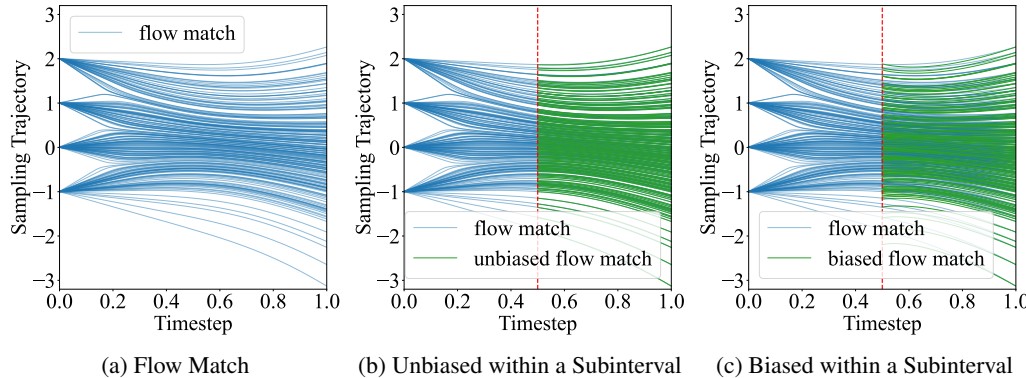

(a) Flow Match      (b) Unbiased within a Subinterval      (c) Biased within a Subinterval

Figure 2: Sampling trajectories for 200 samples in a 1D toy experiment. (a) Training with the full-interval objective (Eq. 4). (b) Training on $0.5 < t < 1$ with the correct subinterval objective (Eq. 13). (c) Training on $0.5 < t < 1$ with an incorrect target: $\|(\boldsymbol{\psi_\theta}(\boldsymbol{x}_t) - (\boldsymbol{\epsilon} - \boldsymbol{x}_s)\|^2$.

encounters singularity and numerical instability. To mitigate this, we apply a clamping function, resulting in the final objective:

$$J_{flow}(\boldsymbol{\theta}) = \mathbb{E}_{\boldsymbol{x}_s \sim p(\boldsymbol{x}_s), \boldsymbol{\epsilon} \sim \mathcal{N}, t \sim \mathcal{T}(t;s,1), \boldsymbol{x}_t = \alpha_{t|s}\boldsymbol{x}_s + \sigma_{t|s}\boldsymbol{\epsilon}}$$

$$[\mathrm{clamp}(1/(\sigma_{t|s})^2)\|\sigma_{t|s}\boldsymbol{\psi_\theta}(\boldsymbol{x}_t) - ((\alpha_s^2\sigma_t + \alpha_t\sigma_s^2)/\alpha_s^2)\boldsymbol{\epsilon} - (\sigma_{t|s}/\alpha_s)\boldsymbol{x}_s)\|^2] \quad (13)$$

Here, $\mathrm{clamp}(1/(\sigma_{t|s})^2)$ restricts the value within a predefined range to prevent overflow.

We design a one-dimensional toy experiment to validate the effect of this training objective, as shown in Fig. 2. The close overlap of the sampling trajectories in Fig. 2b demonstrates that, within the defined subinterval, the flow model trained with Eq. 13 is equivalent to one trained with the standard objective in Eq. 4. Conversely, Fig. 2c illustrates how an incorrect formulation of the objective leads to a biased estimation. Refer detailed settings of toy example to Appendix D.

## 3 EXPERIMENTS AND RESULTS

We apply Phased DMD to state-of-the-art (SOTA) image and video generative models. All experiments are conducted using a 4-step, 2-phase configuration, as illustrated in Fig. 1d. Consequently, each base model is distilled into two expert networks. To demonstrate that the performance improvement stems primarily from our novel distillation paradigm rather than merely an increase in trainable parameters, we include the Wan2.2-T2V-A14B model (Wan et al., 2025) in our experiments. This model already features an MoE structure, and both standard DMD and our Phased DMD distill it into two experts. This allows for a direct comparison under equivalent parameter budgets. Owing to its computational demands, the vanilla DMD (Yin et al., 2024a) method was applied only to the smallest model configuration, namely the Wan2.1-T2V-14B. An overview of the experimental configurations is provided in Tab. 1, with detailed descriptions available in Appendix C.

Table 1: Overview of Experimental Setup.

| Base Model | Task | DMD | DMD with SGTS | Phased DMD (Ours) |
|---|---|---|---|---|
| Wan2.1-T2V-14B | T2I | ✓ | ✓ | ✓ |
| Wan2.2-T2V-A14B | T2I, T2V | ✗ | ✓ | ✓ |
| Qwen-Image-20B | T2I | ✗ | ✓ | ✓ |

## 3.1 PRESERVATION OF GENERATIVE DIVERSITY

To evaluate generative diversity, we constructed a text-to-image test set comprising 21 prompts. Each prompt provides a short description of the image content without detailed specifications. For each prompt, we generated 8 images using seeds from 0 to 7. For the base model, images are sampled using 40 steps with a CFG scale of 4. All distilled models are sampled using 4 steps and a CFG scale of 1. As shown in Fig. 3b, images generated by the 4-step DMD model exhibit a loss of fine details. While the 4-step DMD model with SGTS improves image quality, this comes at the cost of reduced diversity. Fig. 3c reveals that the generated images often adopt a similar close-up view and demonstrate limited variation in composition across different random seeds. In contrast, Phased DMD better preserves diversity, producing images with a wider range of natural compositions, as illustrated in Fig. 3d. Generative diversity is evaluated using two complementary metrics: (1) the mean pairwise cosine similarity of DINOv3 features (Siméoni et al., 2025), where lower values indicate higher diversity, and (2) the mean pairwise LPIPS distance (Zhang et al., 2018), where higher values denote greater diversity. Both metrics are computed across images generated from the same prompt using different seeds. The quantitative results are presented in Tab. 2. As expected, the base models achieve the highest diversity. Notably, DMD with SGTS yields slightly lower diversity than vanilla DMD. Our Phased DMD outperforms both distillation baselines, demonstrating its superior capability for preserving the generative diversity of the original model. The diversity improvement on Qwen-Image is marginal. We argue this stems from the base model's own limited output diversity.

Table 2: Two metrics for quantitative diversity evaluation: average pairwise DINOv3 cosine similarity (lower is better) and LPIPS distance (higher is better). Phased DMD outperforms the vanilla DMD and DMD with SGTS in preserving generative diversity of the base models.

| Method | Wan2.1-T2V-14B | | Wan2.2-T2V-A14B | | Qwen-Image | |
|---|---|---|---|---|---|---|
| | DINOv3 ↓ | LPIPS ↑ | DINOv3 ↓ | LPIPS ↑ | DINOv3 ↓ | LPIPS ↑ |
| Base model | 0.708 | 0.607 | 0.732 | 0.531 | 0.907 | 0.483 |
| DMD | 0.825 | 0.522 | - | - | - | - |
| DMD with SGTS | 0.826 | 0.521 | 0.828 | 0.447 | **0.941** | 0.309 |
| Phased DMD (Ours) | **0.782** | **0.544** | **0.768** | **0.481** | 0.958 | **0.322** |

## 3.2 RETAIN BASE MODELS' KEY CAPABILITIES

Wan2.2 video generation models exhibit remarkable capabilities in motion dynamics and camera control. However, we observe that DMD with SGTS degrade these properties, as they do not specifically address the low-SNR base expert. Phased DMD inherently resolves this issue by dividing distillation into phases and explicitly eliminating dependency on $x_0$ except in the final phase. In the first phase, only the low-SNR expert attends and is distilled according to Eq. 11 and Eq. 13. Since the pre-trained low-SNR expert is also trained on the low-SNR subinterval, this alignment better preserves its capabilities. As shown in Fig. 6, DMD with SGTS generates slower motion dynamics compared to the base model and Phased DMD. Similarly, Fig. 7 show that DMD with SGTS tends to produce close-up views, while Phased DMD and the base model better adhere to the

Table 3: Comparison of motion dynamics preservation across distillation methods, measured by mean absolute optical flow and VBench dynamic degree. Phased DMD demonstrates superior performance in maintaining the base model's motion quality.

| Method | Optical Flow ↑ | Dynamic Degree ↑ |
|---|---|---|
| Base model | 10.66 | 79.35 % |
| DMD with SGTS | 5.27 | 72.90 % |
| Phased DMD(Ours) | **7.76** | **78.71 %** |

Prompt: "A chef meticulously plating a dish."

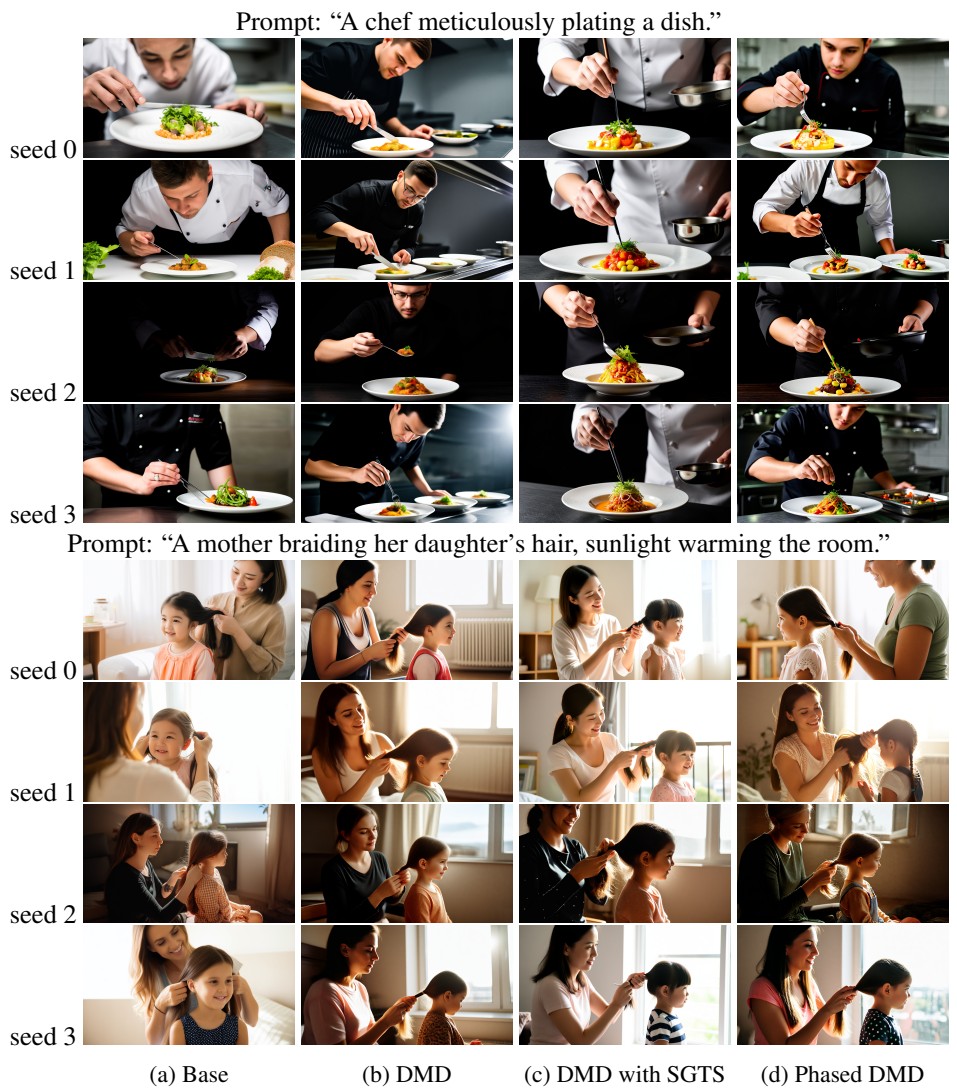

Prompt: "A mother braiding her daughter's hair, sunlight warming the room."

(a) Base       (b) DMD       (c) DMD with SGTS       (d) Phased DMD

Figure 3: Samples (seeds 0-3) from the Wan2.1-T2V-14B base model (40 steps, CFG=4) and its distilled variants (4 steps, CFG=1): (a) Base, (b) DMD, (c) DMD with SGTS, (d) Phased DMD.

prompt's camera instructions. We evaluate motion quality on a set of 155 prompts, generating one video per prompt with a fixed seed $42$. Motion intensity is quantified using the *mean absolute optical flow* computed with Unimatch Xu et al. (2023) and the *dynamic degree* metric from VBench (Zhang et al., 2024). As Tab. 3 shows, Phased DMD produces significantly stronger motion dynamics than DMD with SGTS, confirming its superior ability to preserve the base model's motion capabilities. Additional comparative videos are provided in the supplementary material. We encourage readers to view these videos to better appreciate the contrasts in motion dynamics and camera control.

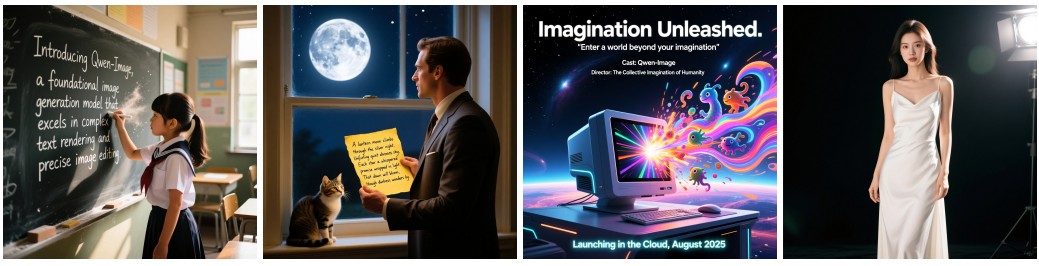

Figure 4: Examples generated by the Qwen-Image distilled with Phased DMD.

Qwen-Image is recognized for its faithful adherence to prompts and high-quality text rendering. To evaluate the preservation of these capabilities after distillation, we applied Phased DMD to Qwen-Image and generated images using prompts from its official website (Team, 2025a). As shown in Fig. 4, the model distilled with Phased DMD exhibits well-preserved capabilities, producing high-quality images with accurate text rendering.

### 3.3 MERIT OF MOE

Our empirical findings reveal that during the distillation process, DMD initially captures structural information before learning finer textural details. Before the complete acquisition of textural details, the generated images and videos tend to exhibit overly smooth features, such as blurry hair and plastic-like skin textures. On the other hand, the mode-seeking nature of reverse KL divergence leads to a decline in generative diversity as training iterations increase. Phased DMD addresses the trade-off between quality and diversity by dividing DMD into distinct training phases. In the low-SNR phases, the composition of images and videos is effectively established. During the subsequent high-SNR phases, the low-SNR expert is frozen, allowing for extended training to enhance generation quality without degrading the structural composition of the outputs. As illustrated in Fig. 5, extending the training of high-SNR experts primarily affects lighting and textural details, while leaving the overall structural composition of the images unchanged.

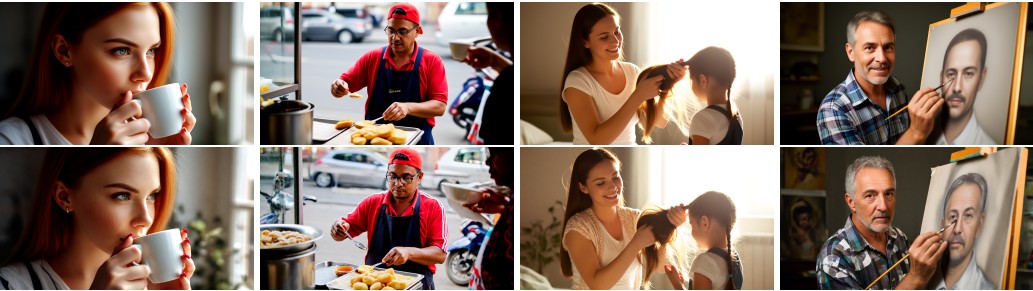

Figure 5: Samples generated with high-SNR experts from different training stages (top: 100 iterations; bottom: 400 iterations) and a shared low-SNR expert. Each column uses identical prompts and seeds.

## 4 RELATED WORKS

Our work builds on Variational Score Distillation (VSD), comprising a trainable generator, a fake score estimator, and a pretrained teacher score estimator. The closest related work is TDM (Luo et al., 2025), which also extends DMD to few-step distillation. Yet, Phased DMD differs in three key ways: (a) TDM lacks theoretical grounding, leading to incorrect fake flow training; (b) our framework inherently produces MoE models; and (c) we use reverse nested SNR intervals, unlike TDM's disjoint intervals. Full discussions about related work are presented in Appendix B.

## 5 CONCLUSION AND DISCUSSION

Phased DMD primarily enhances structural aspects of generation, such as image composition diversity, motion dynamics, and camera control. However, for base models like Qwen-Image, whose outputs are inherently less diverse, the improvement is less pronounced. While this work demonstrates phased distillation within the DMD framework, the approach is generalizable to other objectives like Fisher divergence in SiD (Zhou et al., 2024), which we leave for future exploration. It is conceivable that other methods for enhancing diversity and dynamics, such as incorporating trajectory data pre-generated by the base model, could be integrated. However, this would compromise the data-free advantage central to DMD. While we may explore such directions in the future, this work prioritizes the data-free paradigm.

## 6 ETHICS STATEMENT

This work complies with the ICLR Code of Ethics. The proposed method follows DMD and is a data-free distillation framework. However, the base model used for distillation may generate human figures due to the presence of human data in the training set, potentially raising concerns about privacy and consent. To address this, we focus solely on human motion dynamics, with no use of personally identifiable information. Regarding the video generation model, while it offers positive applications in content creation, it also carries risks of misuse for deceptive content or surveillance. We acknowledge these risks and emphasize that our model is intended strictly for scientific research and positive use cases.

## 7 REPRODUCIBILITY STATEMENT

We have taken extensive measures to ensure reproducibility. To reproduce Phased DMD, the core equations are provided in Sec.2 of the main text, with detailed derivations in Appendix A. For the toy example to verify the effectiveness of score matching with subintervals, relevant details can be found in the Appendix D. To replicate our experiments, details of the experimental setup, hyperparameters, evaluation metircs and implementation choices are available in Sec. 3 of the main text and Appendix 3. Code and models will also be released.

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

## A   DETAILED DERIVATION OF METHOD

We show the detailed derivation of Eq. 5 as follows:

$$J_{flow}(\boldsymbol{\theta}) = \mathbb{E}_{\boldsymbol{x}_0 \sim p(\boldsymbol{x}_0), \boldsymbol{\epsilon} \sim \mathcal{N}, t \sim \mathcal{T}, \boldsymbol{x}_t = \alpha_t \boldsymbol{x}_0 + \sigma_t \boldsymbol{\epsilon}}[\|\boldsymbol{\psi}_{\boldsymbol{\theta}}(\boldsymbol{x}_t) - (\boldsymbol{\epsilon} - \boldsymbol{x}_0)\|^2]$$

$$= \mathbb{E}_{\boldsymbol{x}_0 \sim p(\boldsymbol{x}_0), \boldsymbol{\epsilon} \sim \mathcal{N}, t \sim \mathcal{T}, \boldsymbol{x}_t = \alpha_t \boldsymbol{x}_0 + \sigma_t \boldsymbol{\epsilon}}[\|\boldsymbol{\psi}_{\boldsymbol{\theta}}(\boldsymbol{x}_t) - (\boldsymbol{\epsilon} - (\boldsymbol{x}_t - \sigma_t \boldsymbol{\epsilon})/\alpha_t))\|^2]$$

$$= \mathbb{E}_{\boldsymbol{x}_0 \sim p(\boldsymbol{x}_0), \boldsymbol{\epsilon} \sim \mathcal{N}, t \sim \mathcal{T}, \boldsymbol{x}_t = \alpha_t \boldsymbol{x}_0 + \sigma_t \boldsymbol{\epsilon}}[\|\boldsymbol{\psi}_{\boldsymbol{\theta}}(\boldsymbol{x}_t) + \boldsymbol{x}_t/\alpha_t - (1 + \sigma_t/\alpha_t)\boldsymbol{\epsilon}\|^2]$$

$$= \mathbb{E}_{\boldsymbol{x}_0 \sim p(\boldsymbol{x}_0), t \sim \mathcal{T}, \boldsymbol{x}_t \sim p(\boldsymbol{x}_t|\boldsymbol{x}_0)}[\|\boldsymbol{\psi}_{\boldsymbol{\theta}}(\boldsymbol{x}_t) + \boldsymbol{x}_t/\alpha_t + (\sigma_t + \sigma_t^2/\alpha_t)\nabla_{\boldsymbol{x}_t} \log(p(\boldsymbol{x}_t|\boldsymbol{x}_0))\|^2]$$

$$= \mathbb{E}_{t \sim \mathcal{T}, \boldsymbol{x}_t \sim p(\boldsymbol{x}_t)}[\|\boldsymbol{\psi}_{\boldsymbol{\theta}}(\boldsymbol{x}_t) + \boldsymbol{x}_t/\alpha_t + (\sigma_t + \sigma_t^2/\alpha_t)\nabla_{\boldsymbol{x}_t} \log(p(\boldsymbol{x}_t))\|^2]$$

In the derivation, we use the the score of $p(\boldsymbol{x}_t|\boldsymbol{x}_0)$, i.e., $\nabla_{\boldsymbol{x}_t} \log(p(\boldsymbol{x}_t|\boldsymbol{x}_0)) = -(1/\sigma_t)\boldsymbol{\epsilon}$, and the equivalence between DSM and ESM (Vincent, 2011).

We show the detailed derivation of Eq. 12 as follows:

$$J_{flow}(\boldsymbol{\theta}) = \mathbb{E}_{t \sim \mathcal{T}(t;s,1), \boldsymbol{x}_t \sim p(\boldsymbol{x}_t)}[\|\boldsymbol{\psi}_{\boldsymbol{\theta}}(\boldsymbol{x}_t) + \boldsymbol{x}_t/\alpha_t + (\sigma_t + \sigma_t^2/\alpha_t)\nabla_{\boldsymbol{x}_t} \log(p(\boldsymbol{x}_t))\|^2]$$

$$= \mathbb{E}_{\boldsymbol{x}_s \sim p(\boldsymbol{x}_s), t \sim \mathcal{T}(t;s,1), \boldsymbol{x}_t \sim p(\boldsymbol{x}_t|\boldsymbol{x}_s)}[\|\boldsymbol{\psi}_{\boldsymbol{\theta}}(\boldsymbol{x}_t) + \boldsymbol{x}_t/\alpha_t + (\sigma_t + \sigma_t^2/\alpha_t)\nabla_{\boldsymbol{x}_t} \log(p(\boldsymbol{x}_t|\boldsymbol{x}_s))\|^2]$$

$$= \mathbb{E}_{\boldsymbol{x}_s \sim p(\boldsymbol{x}_s), \boldsymbol{\epsilon} \sim \mathcal{N}, t \sim \mathcal{T}(t;s,1), \boldsymbol{x}_t = \alpha_{t|s} \boldsymbol{x}_s + \sigma_{t|s} \boldsymbol{\epsilon}}[\|\boldsymbol{\psi}_{\boldsymbol{\theta}}(\boldsymbol{x}_t) + \boldsymbol{x}_t/\alpha_t - ((\sigma_t + \sigma_t^2/\alpha_t)/\sigma_{t|s})\boldsymbol{\epsilon}\|^2]$$

$$= \mathbb{E}_{\boldsymbol{x}_s \sim p(\boldsymbol{x}_s), \boldsymbol{\epsilon} \sim \mathcal{N}, t \sim \mathcal{T}(t;s,1), \boldsymbol{x}_t = \alpha_{t|s} \boldsymbol{x}_s + \sigma_{t|s} \boldsymbol{\epsilon}}[\|\boldsymbol{\psi}_{\boldsymbol{\theta}}(\boldsymbol{x}_t) + (\alpha_{t|s} \boldsymbol{x}_s + \sigma_{t|s} \boldsymbol{\epsilon})/\alpha_t - ((\sigma_t + \sigma_t^2/\alpha_t)/\sigma_{t|s})\boldsymbol{\epsilon}\|^2]$$

$$= \mathbb{E}_{\boldsymbol{x}_s \sim p(\boldsymbol{x}_s), \boldsymbol{\epsilon} \sim \mathcal{N}, t \sim \mathcal{T}(t;s,1), \boldsymbol{x}_t = \alpha_{t|s} \boldsymbol{x}_s + \sigma_{t|s} \boldsymbol{\epsilon}}[\|\boldsymbol{\psi}_{\boldsymbol{\theta}}(\boldsymbol{x}_t) - ((\alpha_s^2 \sigma_t + \alpha_t \sigma_s^2)/(\alpha_s^2 \sigma_{t|s}))\boldsymbol{\epsilon} - (1/\alpha_s)\boldsymbol{x}_s)\|^2]$$

The relationship between sample prediction (x-prediction) and score matching is derived as follows:

$$J_{sample}(\boldsymbol{\theta}) = \mathbb{E}_{\boldsymbol{x}_0 \sim p(\boldsymbol{x}_0), \boldsymbol{\epsilon} \sim \mathcal{N}, t \sim \mathcal{T}, \boldsymbol{x}_t = \alpha_t \boldsymbol{x}_0 + \sigma_t \boldsymbol{\epsilon}}[\|\boldsymbol{\mu}_{\boldsymbol{\theta}}(\boldsymbol{x}_t) - \boldsymbol{x}_0\|^2]$$

$$= \mathbb{E}_{\boldsymbol{x}_0 \sim p(\boldsymbol{x}_0), \boldsymbol{\epsilon} \sim \mathcal{N}, t \sim \mathcal{T}, \boldsymbol{x}_t = \alpha_t \boldsymbol{x}_0 + \sigma_t \boldsymbol{\epsilon}}[\|\boldsymbol{\mu}_{\boldsymbol{\theta}}(\boldsymbol{x}_t) - (\boldsymbol{x}_t - \sigma_t \boldsymbol{\epsilon})/\alpha_t)\|^2]$$

$$= \mathbb{E}_{\boldsymbol{x}_0 \sim p(\boldsymbol{x}_0), \boldsymbol{\epsilon} \sim \mathcal{N}, t \sim \mathcal{T}, \boldsymbol{x}_t = \alpha_t \boldsymbol{x}_0 + \sigma_t \boldsymbol{\epsilon}}[\|\boldsymbol{\mu}_{\boldsymbol{\theta}}(\boldsymbol{x}_t) - \boldsymbol{x}_t/\alpha_t + (\sigma_t/\alpha_t)\boldsymbol{\epsilon}\|^2]$$

$$= \mathbb{E}_{\boldsymbol{x}_0 \sim p(\boldsymbol{x}_0), t \sim \mathcal{T}, \boldsymbol{x}_t \sim p(\boldsymbol{x}_t|\boldsymbol{x}_0)}[\|\boldsymbol{\mu}_{\boldsymbol{\theta}}(\boldsymbol{x}_t) - \boldsymbol{x}_t/\alpha_t - (\sigma_t^2/\alpha_t)\nabla_{\boldsymbol{x}_t} \log(p(\boldsymbol{x}_t|\boldsymbol{x}_0))\|^2]$$

$$= \mathbb{E}_{t \sim \mathcal{T}, \boldsymbol{x}_t \sim p(\boldsymbol{x}_t)}[\|\boldsymbol{\mu}_{\boldsymbol{\theta}}(\boldsymbol{x}_t) - \boldsymbol{x}_t/\alpha_t - (\sigma_t^2/\alpha_t)\nabla_{\boldsymbol{x}_t} \log(p(\boldsymbol{x}_t))\|^2] \tag{14}$$

The training objective for x-prediction diffusion models within a subinterval is as follows:

$$J_{sample}(\boldsymbol{\theta}) = \mathbb{E}_{t \sim \mathcal{T}, \boldsymbol{x}_t \sim p(\boldsymbol{x}_t)}[\|\boldsymbol{\mu}_{\boldsymbol{\theta}}(\boldsymbol{x}_t) - \boldsymbol{x}_t/\alpha_t - (\sigma_t^2/\alpha_t)\nabla_{\boldsymbol{x}_t} \log(p(\boldsymbol{x}_t))\|^2]$$

$$= \mathbb{E}_{\boldsymbol{x}_s \sim p(\boldsymbol{x}_s), t \sim \mathcal{T}(t;s,1), \boldsymbol{x}_t \sim p(\boldsymbol{x}_t|\boldsymbol{x}_s)}[\|\boldsymbol{\mu}_{\boldsymbol{\theta}}(\boldsymbol{x}_t) - \boldsymbol{x}_t/\alpha_t - (\sigma_t^2/\alpha_t)\nabla_{\boldsymbol{x}_t} \log(p(\boldsymbol{x}_t|\boldsymbol{x}_s))\|^2]$$

$$= \mathbb{E}_{\boldsymbol{x}_s \sim p(\boldsymbol{x}_s), \boldsymbol{\epsilon} \sim \mathcal{N}, t \sim \mathcal{T}(t;s,1), \boldsymbol{x}_t = \alpha_{t|s} \boldsymbol{x}_s + \sigma_{t|s} \boldsymbol{\epsilon}}[\|\boldsymbol{\mu}_{\boldsymbol{\theta}}(\boldsymbol{x}_t) - \boldsymbol{x}_t/\alpha_t + ((\sigma_t^2/\alpha_t)/\sigma_{t|s})\boldsymbol{\epsilon}\|^2]$$

$$= \mathbb{E}_{\boldsymbol{x}_s \sim p(\boldsymbol{x}_s), \boldsymbol{\epsilon} \sim \mathcal{N}, t \sim \mathcal{T}(t;s,1), \boldsymbol{x}_t = \alpha_{t|s} \boldsymbol{x}_s + \sigma_{t|s} \boldsymbol{\epsilon}}[\|\boldsymbol{\mu}_{\boldsymbol{\theta}}(\boldsymbol{x}_t) - (\alpha_{t|s} \boldsymbol{x}_s + \sigma_{t|s} \boldsymbol{\epsilon})/\alpha_t + ((\sigma_t^2/(\alpha_t \sigma_{t|s})\boldsymbol{\epsilon}\|^2]$$

$$= \mathbb{E}_{\boldsymbol{x}_s \sim p(\boldsymbol{x}_s), \boldsymbol{\epsilon} \sim \mathcal{N}, t \sim \mathcal{T}(t;s,1), \boldsymbol{x}_t = \alpha_{t|s} \boldsymbol{x}_s + \sigma_{t|s} \boldsymbol{\epsilon}}[\|\boldsymbol{\mu}_{\boldsymbol{\theta}}(\boldsymbol{x}_t) - ((1/\alpha_s)\boldsymbol{x}_s - (\alpha_t \sigma_s^2/\alpha_s^2 \sigma_{t|s})\boldsymbol{\epsilon})\|^2] \tag{15}$$

Optimizing within the subinterval according to Eq. 15 gives an unbiased estimation of x-prediction. In contrast, the objective $[\|\boldsymbol{\mu}_{\boldsymbol{\theta}}(\boldsymbol{x}_t) - \boldsymbol{x}_s\|^2]$ yields a biased estimation.

## B   RELATED WORKS

Our work is situated within the framework of Variational Score Distillation (VSD). VSD involves three components: a trainable generator, a fake score estimator, and a pretrained teacher score estimator. The generator is optimized to produce a distribution that approximates the real data distribution. Concurrently, the fake score estimator learns to estimate the score of the generator's output distribution. The update direction for the generator is then determined by the discrepancy between the teacher's score (for the real distribution) and the fake score estimator's score.

Similar to GANs, the VSD framework is adversarial. The fake score estimator must be precisely optimized to learn the score of the current generated distribution. This accurate estimation is crucial, as it combines with the fixed teacher model (which provides the score for the real data) to produce

a correct guidance signal for the generator. This principle explains why DMD2 (Yin et al., 2024a) operates successfully without external real data, in contrast to its predecessor DMD (Yin et al., 2024b).

A key advantage of VSD over GANs for distilling pre-trained diffusion models is initialization. The pre-trained model serves a dual role: it is a powerful multi-step generator and an accurate estimator of the real data distribution's score. This allows it to effectively initialize all three components in the VSD framework, leading to significantly enhanced training stability.

Several methods are built upon the VSD framework, including Diff-Instruct (Luo et al., 2023), DMD (Yin et al., 2024a), SID (Zhou et al., 2024), and FGM (Huang et al., 2024). The fundamental distinction between these approaches lies in the specific divergence they minimize. DMD, for instance, optimizes the reverse KL divergence between the real and generated distributions. A key advantage of this choice is its computational efficiency compared to alternatives like the Fisher divergence used in SID (Zhou et al., 2024). Specifically, during generator optimization, DMD does not require gradients to be backpropagated through the fake and teacher score estimators, whereas SID does. This does not imply the two estimators are trainable in this stage for SID, but rather reflects a difference in the computational graph. This property makes DMD more amenable to engineering implementation and scalable to large base models.

Similar to our work, TDM (Luo et al., 2025) also aimed to extend DMD to few-step distillation. However, our approach differs from TDM in three key aspects: (a) The lack of proper theoretical grounding in TDM renders its fake flow training formulation incorrect, undermining the foundations of DMD. (b) Our framework inherently produces MoE models for few-step generation. (c) While TDM uses disjoint SNR intervals, our method employs reverse nested intervals, where each interval is a subset of the subsequent one.

## C  EXPERIMENTAL DETAILS

We conduct experiments on two tasks: text-to-image and text-to-video generation. The following global settings are applied across all experiments: a batch size of 64; a fake diffusion model learning rate of 4e-7 with full-parameter training; a generator learning rate of 5e-5 using LoRA with a rank of 64 and an alpha value of 8. The AdamW optimizer is used for both the fake diffusion model and the generator, with hyperparameter $\beta_1 = 0, \beta_2 = 0.999$. The fake diffusion model is updated five times for every generator update.

For the Wan2.x base models, distillation for the text-to-image task is performed at a data resolution of $\text{frame} = 1, \text{width} = 1280, \text{height} = 720$.

For the Wan2.2-T2V-A14B model, distillation for the text-to-video task uses a mixture of data resolutions: $(81, 720, 1280), (81, 1280, 720), (81, 480, 832), (81, 832, 480)$.

For the Qwen-Image model, distillation for the text-to-image task uses a mixture of data resolutions: $(1, 1382, 1382), (1, 1664, 928), (1, 928, 1664), (1, 1472, 1104), (1, 1104, 1472), (1, 1584, 1056), (1, 1056, 1584)$.

## D  TOY EXAMPLE DETAILS

We construct a toy example where $x_0$ takes only four values: {-1, 0, 1, 2}. A minimal model is designed, consisting of four MLPs with dim=512, conditioned solely on $t$. Fig. 1a shows training on the full interval using Eq. 4, Fig. 1b shows training on subintervals using Eq. 13, and Fig. 1c shows training on subintervals using Eq. 4, simply replacing $x_0$ with $x_s$. As shown in Fig. 4b, when the correct objective is used, the trajectories on subintervals perfectly align with those on the full interval. In contrast, using an incorrect objective introduces trajectory deviations, as illustrated in Fig. 4c. Such a trajectory deviation signifies that the trained model no longer satisfies the score-matching objective (*i.e.*, Eq. 5 is violated), thus contravening a core principle of DMD.

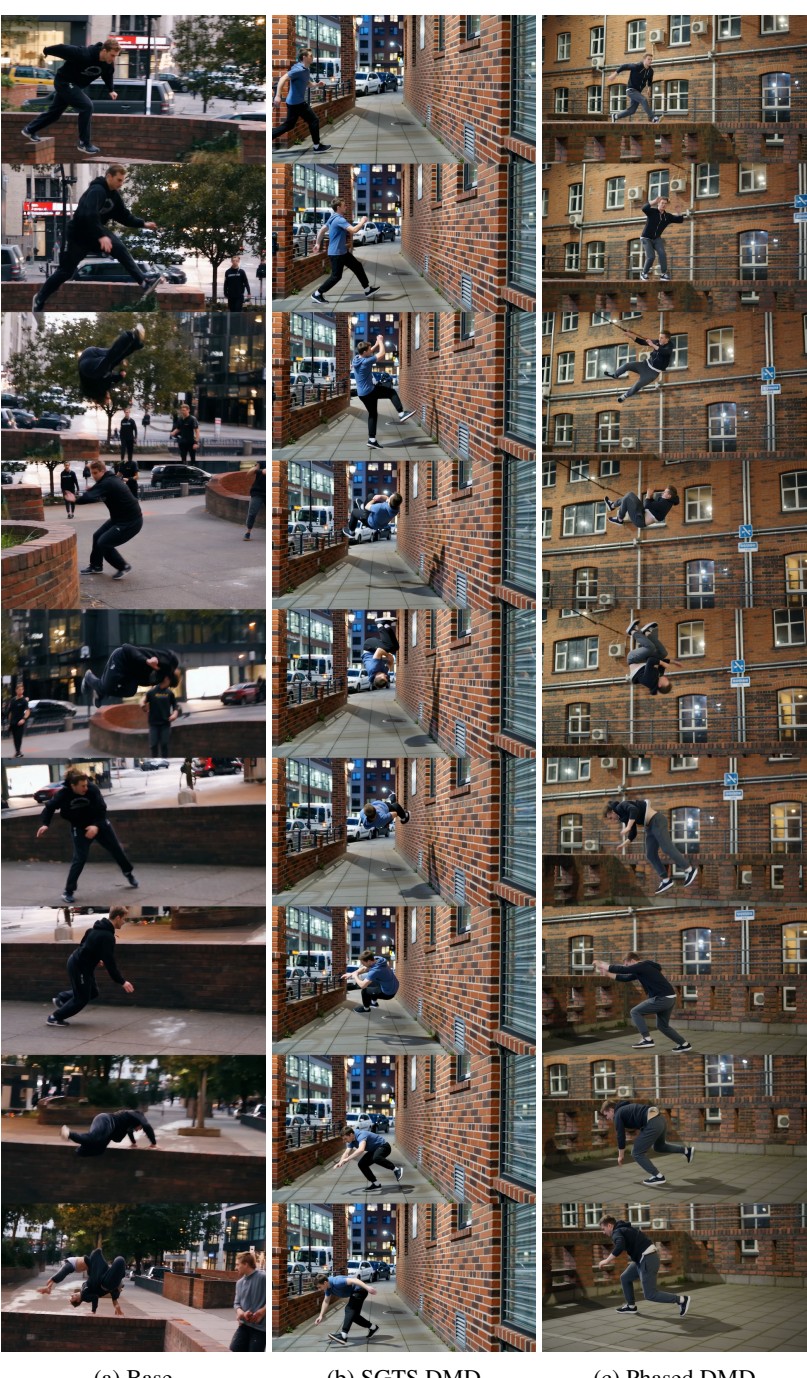

(a) Base        (b) SGTS DMD        (c) Phased DMD

Figure 6: Comparison of video frames generated by the Wan2.2-T2V-A14B base model and its distilled versions using DMD with SGTS and Phased DMD. Each video consists of 81 frames and frames with indices $\{0, 10, ..., 80\}$ are combined as a preview. The base model was sampled with 40 steps and CFG of 4, while the distilled models used 4 steps and CFG of 1 (seed fixed at 42). The prompt is "A parkour athlete swiftly runs horizontally along a brick wall in an urban setting. Pushing off powerfully with one foot, they launch themselves explosively into a twisting front flip. The camera tenaciously stays with them in mid-air as they tuck their legs tightly to their chest to rapidly accelerate the rotation, then extend them forcefully outwards again, precisely spotting their landing on the concrete below. The dynamic movement is vividly captured against a backdrop of city lights and shadows."

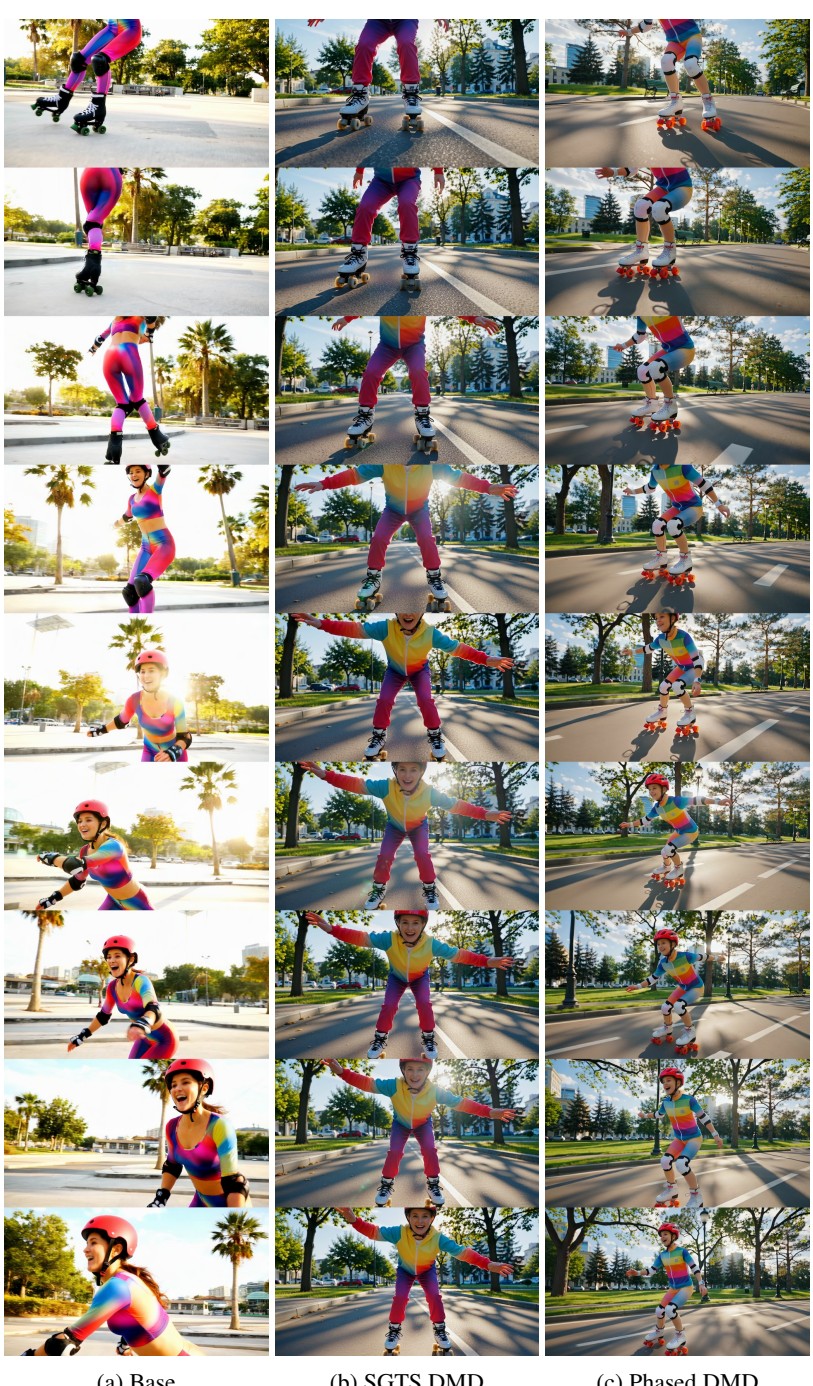

(a) Base        (b) SGTS DMD        (c) Phased DMD

Figure 7: Comparison of video frames generated by the Wan2.2-T2V-A14B base model and its distilled versions using DMD with SGTS and Phased DMD. Each video consists of 81 frames and frames with indices $\{0, 10, ..., 80\}$ are combined as a preview. The base model was sampled with 40 steps and CFG of 4, while the distilled models used 4 steps and CFG of 1 (seed fixed at 42). The prompt is "Day time, sunny lighting, low angle shot, warm colors. A dynamic individual in a vibrant, multi-colored outfit and a red helmet executes a fast-paced slalom on roller skates through a bustling urban park. The camera starts focused on the skates carving sharp turns on the pavement and tilts up to reveal their entire body leaning into the motion. Their face shows a mix of joy and deep concentration. The warm afternoon sun filters through the lush greenery, with the azure sky visible above, creating a scene bursting with energy."

