# OpenReview forum: "Phased DMD: Few-step Distribution Matching Distillation via Score Matching within Subintervals"
_ICLR.cc/2026/Conference — ICLR 2026 Conference Withdrawn Submission_

### Official Review · Reviewer_PJDq · 2025-10-26

**Soundness:** 2
**Presentation:** 2
**Contribution:** 2
**Rating:** 4
**Confidence:** 4

**Summary:**

This work presents Phased DMD. Motivated by the task differences that arise across signal‑to‑noise ratio (SNR) ranges in score‑based models, Phased DMD splits the SNR range into subintervals and performs few‑step DMD progressively, phase by phase. From an architectural perspective, this work also proposes using a Mixture‑of‑Experts (MoE) model where each expert is responsible for a specific SNR subinterval.

With this design, the proposed method demonstrates better diversity than both the original DMD and the stochastic gradient truncation strategy introduced in Self-Forcing, in a multi‑step distillation setup for image generation.

**Strengths:**

- While the idea of progressive diffusion distillation under various criteria has been explored in previous studies such as [1, 2], the specific idea, splitting the SNR range into sub‑intervals to perform progressive multi‑step DMD coupled with MoE, is simple, novel, and interesting.

- In addition, Section 2.3.2 derives an objective that theoretically enables score distillation within sub‑intervals, which is another strength.

- Although the experimental evaluation is not comprehensive, the main body and supplementary materials suggest that the method successfully distills extremely large-scale models (20B and 28B parameters) for image and video generation.

[1] Tim Salimans, et al. "Progressive distillation for fast sampling of diffusion models." ICLR 2022

[2] Dongjun Kim, et al. "PaGoDA: Progressive Growing of a One-Step Generator from a Low-Resolution Diffusion Teacher."  NeurIPS 2024

**Weaknesses:**

Although the proposed method is new and interesting, the major weakness of this work as a scientific paper is the insufficient experimental evaluation. For instance, diversity is evaluated only for image generation (without video generation), and the metrics (DINOv3 and LPIPS) used do not seem to be standard for image generation. For video generation, the evaluation is limited to optical flow, dynamic degree, and screenshots of generated samples.

To comprehensively assess the effectiveness of the proposed method, the reviewer suggests conducting a thorough evaluation using the standard benchmarks and metrics from [3–6] including subjective evaluations.

[3] Xun Huang, et al. "Self Forcing: Bridging the Train-Test Gap in Autoregressive Video Diffusion." NeurIPS 2025

[4] Shanchuan Lin, et al. "Diffusion adversarial post-training for one-step video generation." arXiv preprint arXiv:2501.08316 (2025).

[5] Wan, Team, et al. "Wan: Open and advanced large-scale video generative models." arXiv preprint arXiv:2503.20314 (2025).

[6] Seawead, Team, et al. "Seaweed-7b: Cost-effective training of video generation foundation model." arXiv preprint arXiv:2504.08685 (2025).

**Questions:**

Could you elaborate on the training details? There seems to be a significant amount of missing information regarding reproducibility. For example:
- $t_k$
- text prompts for training and evaluation
- the clamping value
- training iterations
etc...

The proposed method is interesting and convincing, so if its effectiveness can be sufficiently demonstrated through additional comprehensive experiments, the reviewer would be willing to raise the rating toward 'accept'.

---

### Official Review · Reviewer_sMYJ · 2025-10-27

**Soundness:** 2
**Presentation:** 2
**Contribution:** 2
**Rating:** 2
**Confidence:** 4

**Summary:**

This research proposes an incremental version of Distribution Matching Distillation (DMD) by dividing the signal-to-noise ratio (SNR) range into multiple subintervals (“phases”) and performing phase-wise score matching within each. The authors claim that this approach increases training stability and preserves generative diversity in both image and video generation tasks.

**Strengths:**

- The phased, SNR-subinterval approach provides a theoretically grounded extension to DMD.
- This research improves the diversity of image and video distillation based on the extensive experimental results.
- The experimental results are tested with large models, which proves the scalability.
- From my perspective, the paper is easy to read and follow as the mathematical formulas are neat.

**Weaknesses:**

Although the research improves the diversity of distilled results, as claimed, the paper still contains several drawbacks.
- The paper claims that the proposed approach can improve the diversity of generation. However, the theoretical connection between phase-based SNR learning is weak. The improved results might come from the larger capacity of a mixture of experts, which can store more information, but not phase learning.
- Although the paper claims that the huge computation cost and large amount of memory needed for a mixture of experts can be manageable by using LORAs; however, the actual training cost increase from multi-phase distillation is not clearly quantified.
- The experiment needs to be more extensive. Some more analysis can be added to the research; a different number of phases used for obtaining generators is an important factor for readers.
- Some ablation tests should be included for a better understanding of the proposed method.

**Questions:**

The biggest question is about the theoretical connection between phase learning and the improvement in the diversity of generation. Otherwise, please see the above weaknesses.

---

### Official Review · Reviewer_B184 · 2025-10-31

**Soundness:** 2
**Presentation:** 2
**Contribution:** 2
**Rating:** 4
**Confidence:** 3

**Summary:**

The paper proposes a novel approach, called Phased DMD, for distillation of diffusion models into the few-step generators. The authors propose to formulate the few-step generation process in a manner of MoE (Mixture of Experts): their model generates a sequence of progressively less noisy samples $x_{t_i}$, where the translation $x_{t_{i - 1}} \to x_{t_i}$ is performed by the i-th trainable expert $G_{\phi_i}$. The experts (parameterized by LoRAs) are trained in the curriculum from lower to higher SNR (signal-to-noise ratio) along with the (one, fully trainable) corresponding fake score model, which utilizes score matching on the subintervals $(t_{i}, 1)$. The proposed Phased DMD algorithm demonstrates superior generation quality compared to the "vanilla DMD" without losing diversity, opposed to the SGTS baseline.

**Strengths:**

1) The proposed method demonstrates increased generation quality and sample diversity compared to the baselines;
2) Applicability of the method is demonstrated in high-dimensional settings corresponding to the state-of-the-art image and video models.

**Weaknesses:**

### Positioning and contributions

1) First, the score matching within subintervals, proposed as one of the key contributions, is not novel. It is a straightforward consequence of the general score identity $\nabla_y \log p_{Y}(y) = \int \nabla_y \log p_{Y | X}(y | x) p_{X | Y}(x | y) d x$, deeply discussed in e.g. [1]. Similar formulation was applied for the subsequent discrete timesteps in DSB [2]. The exact same (as in Phased DMD) subinterval formulation was applied in e.g. [3];
2) I think there is a significant misunderstanding of the DMD2 paper, referred to as vanilla DMD in the manuscript. The authors state that DMD2 performs backpropagation through the few-step generation process. On the other hand, to my knowledge, DMD2 does not propagate gradients through any of the generation steps except the last, thus treating the intermediate samples $x_{t_i}$ as the synthetic (but "detached") data, used to tackle the typical mismatch between the input distributions at training and inference. DMD2 thus seems to fit the SGTS scheme, shown in Figure 1(b).

### Experiments
3) The paper lacks baselines other than simulation-based DMD and DMD with SGTS;
4) There is almost no quantitative evalutation of the method on the image generation tasks (except for the sample diversity in Table 2);

### Writing quality
5) I think the writing has both overcomplicated notions (with such notations as $\epsilon \sim \mathcal{N}, x_{t_k} = \text{pipeline}(G_{\phi_1}, \ldots, G_{\phi_k}, \{t_1, \ldots, t_k\}, \epsilon, \mathcal{S}), t \sim \mathcal{T}(t; t_k, 1), x_t \sim p(x_t | x_{t_k})$ under expectation) and underexplained notions, which significantly harms readability. For example, how are the scheduler $\mathcal{S}$ and the pipeline implemented in practice? Does the $i$-th expert deterministically generate the next image with higher SNR, or predict the clean image and adds independent noise like it was done in DMD2? What is the parameterization of the few-step generator: clean prediction, $v$-prediction, or something else? Division of the resulting pipeline into few-step (rather than one-step) sub-phases is also underexplained;
6) The manuscript contains several notational inaccuracies such as $\nabla x_t$ instead of $\nabla_{x_t}$ or defining the diffusion model as $F_\theta(x_t)$ without conditioning on the corresponding time step.

[1] Target Score Matching

[2] Diffusion Schrödinger Bridge with Applications to Score-Based Generative Modeling

[3] A Flexible Diffusion Model

**Questions:**

Please see weaknesses.

---

### Note · Authors · 2025-11-14

I have read and agree with the venue's withdrawal policy on behalf of myself and my co-authors.